# 🤖 SE-GUI: Enhancing Visual Grounding for GUI Agents via Self-Evolutionary Reinforcement Learning

**Xinbin Yuan**[1,2*]   **Jian Zhang**[2‡]   **Kaixin Li**[3]   **Zhuoxuan Cai**[2]   **Lujian Yao**[2]   **Jie Chen**[2]
**Enguang Wang**[2]   **Qibin Hou**[1✉]   **Jinwei Chen**[2]   **Peng-Tao Jiang**[2]   **Bo Li**[2✉]

[1]VCIP, School of Computer Science, NKU    [2]vivo Mobile Communication Co., Ltd
[3] National University of Singapore
*yxb@mail.nankai.edu.cn, houqb@nankai.edu.cn, libra@vivo.com*
Project page: `https://github.com/YXB-NKU/SE-GUI`

## Abstract

Graphical User Interface (GUI) agents have made substantial strides in understanding and executing user instructions across diverse platforms. Yet, grounding these instructions to precise interface elements remains challenging—especially in complex, high-resolution, professional environments. Traditional supervised fine-tuning (SFT) methods often require large volumes of diverse data and exhibit weak generalization. To overcome these limitations, we introduce a reinforcement learning (RL)-based framework that incorporates three core strategies: (1) seed data curation to ensure high-quality training samples, (2) a dense policy gradient that provides continuous feedback based on prediction accuracy, and (3) a self-evolutionary reinforcement finetuning mechanism that iteratively refines the model using attention maps. With only 3k training samples, our 7B-parameter model achieves state-of-the-art results among similarly sized models on three grounding benchmarks. Notably, it attains 47.3% accuracy on the ScreenSpot-Pro dataset—outperforming much larger models, such as UI-TARS-72B, by a margin of 24.2%. These findings underscore the effectiveness of RL-based approaches in enhancing GUI agent performance, particularly in high-resolution, complex environments.

## 1   Introduction

Graphical User Interface (GUI) agents have become increasingly capable of executing user commands across diverse platforms [1; 2]. Yet, a core challenge remains: accurately grounding natural language instructions to the correct elements on the interface [3; 4; 5]. While Supervised Fine-Tuning (SFT) has been proven to be effective in simple scenarios [6; 7; 8], it faces two major limitations: requiring large volumes of diverse data and exhibiting weak generalization in complex and high-resolution professional settings [9; 10].

To address these issues, reinforcement learning (RL) offers a promising alternative. Unlike SFT, RL enables models to learn from structured and incremental feedback via reward functions, guiding them toward more precise grounding. Among RL-based methods [11; 12; 13], Group Relative Policy Optimization (GRPO) [13] stands out for its efficiency. It replaces heavy value models with a simple

---

*This work was done during Xinbin Yuan's internship at vivo.
‡Project lead.
✉Corresponding author.

39th Conference on Neural Information Processing Systems (NeurIPS 2025).

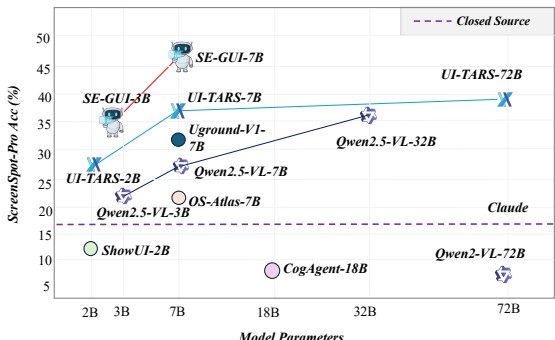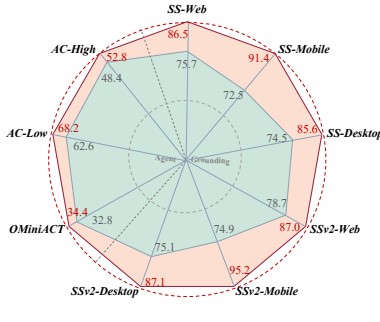

Figure 1: Performance comparison of various GUI agents on both the grounding benchmark (ScreenSpot [18], ScreenSpot-v2 [7], ScreenSpot-Pro [4]) and the agent benchmark (Android_Control-High [19], Android_Control-Low [19], OmniACT [20]). Our model, SE-GUI marked with a star, demonstrates state-of-the-art performance against models with larger parameter counts.

rule-based reward system, reducing the computational burden. However, typical approaches [9; 10] rely on binary (0/1) rewards, which are often sparse. Such sparsity impedes early-stage training, as incorrect predictions commonly receive identical zero rewards, resulting in uniform advantage estimates and limited gradient information for effective optimization.

Furthermore, recent studies [14; 15; 16] also emphasize the importance of training data quality for RL-based methods, which is not explored for GUI tasks. However, grounding datasets are typically collected through automated pipelines that extract data from noisy UI accessibility trees or raw HTML, often without thorough verification [7; 6; 8; 17]. This process introduces significant noise, including low-quality instructions (e.g., generic labels like <PushButton>) and bounding boxes corresponding to elements that appear in the DOM but are not visually rendered in the UI. This raises a critical question: *How can we assess the quality of each training sample?*

To investigate the above problem, we analyze model failures by examining layer attention across the transformer architecture. We found two common patterns: In some cases, the model roughly identifies the target region, but fails to localize it precisely; in others, it completely overlooks the relevant area. These errors often stem from weak alignment between the instruction and the ground truth location.

Motivated by these insights, we propose a Self-Evolutionary reinforcement fine-tuning algorithm that leverages layer attention from a trained model to guide further learning. As grounding accuracy improves, so does the quality of attention, enabling the model to iteratively supervise its own training. At each stage, the best-performing model generates attention maps that inform the next iteration, continuing until performance converges. We summarize our contributions as follows:

1. Seed Data Curation. We curate a 3,018-sample dataset by filtering out vague, inaccurate, or overly simple tasks from a larger candidate pool. This ensures linguistic consistency and balanced task complexity, promoting better generalization and stable performance across scenarios.

2. Group Relative Policy Optimization with Dense Point Reward. To combat sparse rewards, we designed a continuous reward mechanism that evaluates the proximity between predictions and ground truth. This provides smoother feedback, enabling the model to learn from near-misses and gradually refine its grounding behavior.

3. Self-Evolutionary Reinforcement Fine-Tuning. We implement an iterative learning loop, where attention maps serve as intermediate supervision signals. These maps highlight which visual tokens the model attends to for each instruction, helping align its focus with relevant interface elements over time.

We evaluate our SE-GUI on six diverse grounding and agent benchmarks across desktop, mobile, and web environments. As illustrated in Fig. 1, SE-GUI achieves state-of-the-art performance, most notably attaining 47.3% accuracy on the challenging ScreenSpot-Pro benchmark, surpassing the previous best (UI-TARS-72B) by 24.2% at 7B parameter scale with only 3k training samples. These

results demonstrate the effectiveness of our RL-based approach for grounding in complex GUI environments.

## 2 Related Work

### 2.1 GUI Agents

GUI agents, as a specialized class of autonomous AI systems, have rapidly evolved with the advent of large foundation models such as LLMs and vision-language models (VLMs) [21; 22; 23], enabling them to interact with graphical user interfaces in increasingly sophisticated ways. Unlike traditional programmatic agents that rely on API calls or internal code access, GUI agents [24; 25; 26; 27; 28] simulate human-like interactions through mouse clicks, keyboard inputs, and visual perception, offering greater flexibility in automating tasks across diverse platforms. Early approaches [29] focused on structured representations like HTML code, but recent advancements [6; 8; 7] demonstrate superior performance when agents directly process visual forms of GUIs, leveraging high-resolution encoders and unified vision-language interfaces . Systems like AppAgent [24; 25], and CogAgent [26] have pioneered this field by enhancing GUI comprehension and interaction precision, while newer frameworks such as UI-TARS [30] and AgentS2 [3] introduce modular architectures and generalist-specialist designs for improved task planning and cross-platform generalization. Despite these strides, challenges remain, particularly in grounding accuracy and data efficiency, as highlighted by methods like UGround [6] and OS-Atlas [7]. However, the reliance on supervised fine-tuning (SFT) poses limitations in scalability and adaptability, motivating the exploration of advanced learning paradigms to overcome the need for vast labeled datasets and improve generalization to unseen interfaces.

### 2.2 Reinforcement Learning

Reinforcement Learning (RL) has emerged as a transformative approach in the domain of GUI agents [9; 10], leveraging rule-based reward functions to guide model behavior and enhance performance. Frameworks like DeepSeek-R1 [23] have demonstrated the efficacy of RL in tasks such as mathematical reasoning and code generation, with subsequent studies [31; 32] extending its application to multimodal models for visual tasks, including image classification and object detection. as to the application of Reinforcement Fine-Tuning (RFT) in gui tasks. UI-R1 [9] and GUI-R1 [10] represents a pioneering effort in this direction, showcasing the potential of RFT to improve action prediction accuracy and grounding in GUI environments, even with limited data. These developments highlight RL's adaptability and scalability, positioning it as a promising paradigm for future innovations in intelligent GUI agent. However, these approaches typically rely on binary (0/1) rewards, which are often sparse. Such sparsity impedes early-stage training, as incorrect predictions commonly receive identical zero rewards, resulting in uniform advantage estimates and limited gradient information for effective optimization. So, how to design better RL algorithms remains a question worth exploring.

## 3 Method

In this section, we describe our proposed framework SE-GUI. It comprises three key components as shown in Fig. 2. First, we introduce *a data filtering strategy* aimed at constructing a clean training set, addressing the need for high-quality data in RL. Second, to mitigate the problem of sparse binary reward signals, we introduce *a dense point reward mechanism* to guide the model more effectively towards the target even when initial predictions are imprecise. Finally, we propose *a self-evolutionary training strategy* that iteratively refines the model and data with an attention-based filtering method.

### 3.1 Seed Data Curation

Our work commences with the collection of diverse open-source grounding datasets including ShowUI [33], UGround [6] and AriaUI [17]. This leads to approximately 300k examples of grounding data encompassing desktop, web, and mobile platforms. These datasets are primarily created through automated pipelines, which often introduce noise, such as inaccurate or off-screen bounding boxes. To address these issues and ensure high training quality, we implemented a three-fold filtering strategy to construct a cleaner and more reliable seed dataset:

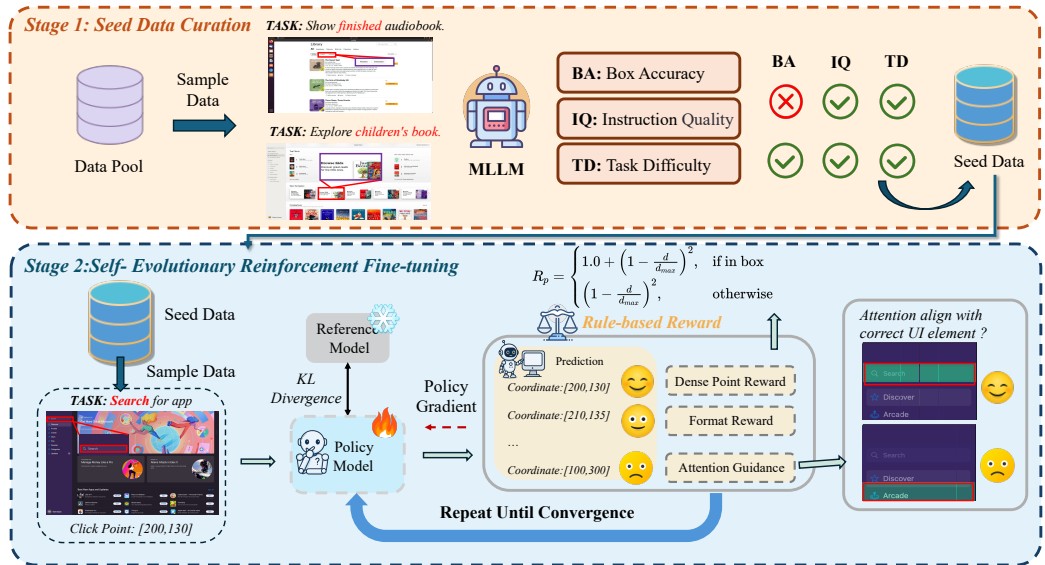

Figure 2: **Overview of the SE-GUI Framework.** Given high-level instructions and the corresponding screenshots, SE-GUI generates multiple candidate responses and is optimized via GRPO using verifiable rewards as feedback signals. The updated model is then leveraged to identify high-quality samples in the training set, which yields cleaner learning signal for subsequent iterations. This iterative process enables SE-GUI to progressively improve itself by refining its predictions and training data.

**Instruction quality.** We use regular expressions to remove low-quality entries such as raw python object names. We then employ a multi-modal large language model to score the remaining samples(The prompt is provided in appendix), evaluating the clarity of the instruction, its alignment with the UI elements, and potential ambiguity.

**Bounding box accuracy.** Many original annotations are misaligned or include irrelevant components. To correct this, we train a bounding box quality scorer based on the Qwen2.5-VL-7B model. This scorer is fine-tuned on a curated set of high-quality bounding boxes from our collected 100k web pages as positive examples and randomly positioned ones as negatives, allowing it to effectively filter out inaccurate annotations.

**Task difficulty.** We utilize the Qwen2.5-VL-7B base model to conduct zero-shot testing on each data point, generating 8 output results for each instruction. If all results are correct, the data point is considered too simple and filtered out.

After applying these three filtering steps, we curated a final set of 3,018 high-quality samples, which we refer to as SE-GUI-3k.

## 3.2 Group Relative Policy Optimization with Dense Point Reward

In our RL training process, the model generates a set of $N$ potential position predictions $\{o_1, o_2, \ldots, o_N\}$. Each response is evaluated by format reward and point reward $\{r_1, r_2, \ldots, r_N\}$. These rewards are then normalized to calculate the relative advantage of each response. The relative quality $A_i$ of the $i$-th response is computed as

$$A_i = \frac{r_i - \text{Mean}(\{r_1, r_2, \ldots, r_N\})}{\text{Std}(\{r_1, r_2, \ldots, r_N\})}, \tag{1}$$

where $\text{Mean}$ and $\text{Std}$ are the mean and standard deviation of the rewards. This normalization step ensures that responses are compared within the context of the group, allowing model to better capture nuanced differences between candidates. Policy updates are further constrained by minimizing the KL divergence between the updated and reference models, ensuring stable RL learning.

However, during early-stage training, incorrect predictions are common. If we rely solely on binary rewards (e.g., success or failure), the resulting advantage estimates tend to be uniform, providing

limited gradient signals for effective learning. To address the issue of sparse reward feedback, we introduce a dense point reward mechanism. This reward $R_p$ is calculated by comparing the predicted click point $(x, y)$ with the ground truth bounding box $gt^{\text{bbox}} = [x_1, y_1, x_2, y_2]$. The calculation formula is as follows:

$$d = \sqrt{\left(\frac{x}{W} - \frac{x_1 + x_2}{2 \cdot W}\right)^2 + \left(\frac{y}{H} - \frac{y_1 + y_2}{2 \cdot H}\right)^2}, \tag{2}$$

$$R_{\text{p}} = \begin{cases} 1.0 + \left(1 - \frac{d}{d_{\text{max}}}\right)^2, & \text{if } \begin{cases} x_1 \leq x \leq x_2, \\ y_1 \leq y \leq y_2 \end{cases} \\ \left(1 - \frac{d}{d_{\text{max}}}\right)^2, & \text{otherwise} \end{cases} \tag{3}$$

where $H$ and $W$ are image width and height, respectively, d means the normalized distance between the click point and the center point of the ground truth bounding box and $d_{max}$ is the normalized maximum distance between the center point of the ground truth bounding box and the four vertices of the image. Essentially, this formula assigns a reward $R_p$ based on whether a predicted point $(x, y)$ lies within a bounding box and its normalized distance from the box's center. If the point is inside the box (i.e., $x_1 \leq x \leq x_2$ and $y_1 \leq y \leq y_2$), the base reward is $1.0$, and an additional decay term reduces the reward as the point moves farther from the center. If the point is outside the box, the reward only depends on the normalized distance between the point and the box center.

Finally, following previous works [34; 23; 32], we introduce format rewards during training to evaluate whether the generated output adheres to the expected output format. The final response reward is composed of format rewards and point rewards, defined as: $R_o = \alpha R_f + \beta R_{\text{Point}}$, where $R_f$ represents the format reward, $R_{\text{Point}}$ represents the point reward, and $\alpha$ and $\beta$ are weighting parameters, respectively.

### 3.3 Self-Evolutionary Reinforcement Fine-Tuning

The proposed algorithm, SE-RFT (Self-Evolutionary Reinforcement Fine-Tuning), outlines a principled approach to iteratively improving a vision-language GUI agent through attention-guided self-supervision. We begin by training an initial model using seed data. In each subsequent training iteration, the model from the previous round is used to generate attention maps for the current training samples. To do this, we generate the model's output sequence and collect self-attention weights from the decoder across all transformer layers. Subsequently, the attention vector for each generated token is normalized, retaining only the components related to the visual tokens. Next, an average is computed across all layers to obtain an aggregated token-to-vision attention weight matrix. Finally, these attention values are projected back onto the original image resolution to produce spatial attention maps. These attention maps highlight the visual tokens on which the current instruction focuses.

If an attention map fails to focus appropriately on the correct UI element, it may indicate that *the model lacks relevant prior knowledge or that the sample presents a challenge beyond the model's current capacity*. We found that such cases frequently lead to inaccurate localization.

Taking this into account, we propose to guide the loss function with the attention maps. If attention maps cannot correctly attend to the target UI element, these data may lead to incorrect guidance during training, hindering model optimization. To address this, we set the loss for these data points to zero, ensuring that they do not negatively impact the training process. The formula below is used to compute whether the attention map is devoted to the correct target UI element.

$$f(\text{attn}, gt^{\text{bbox}}, \tau) = \begin{cases} 1, & \text{if } P_{\text{peak}} \wedge P_{\text{global}} \\ 0, & \text{otherwise} \end{cases} \tag{4}$$

where $attn$ means the spatial attention maps, $\tau$ is the filtering threshold, $P_{\text{peak}}$ indicates if there exists any significant activation point in the target region:

$$P_{\text{peak}} = \mathbf{1}\left(\max_{(i,j) \in [x_1, x_2] \times [y_1, y_2]} \text{attn}[i, j] > \tau\right), \tag{5}$$

and $P_{\text{global}}$ indicates if the average attention weight in the target region is higher than the global average:

$$P_{\text{global}} = \mathbf{1} \left( \frac{1}{H_{\text{gt}} W_{\text{gt}}} \sum_{i=x_1}^{x_2} \sum_{j=y_1}^{y_2} \text{attn}[i,j] > \frac{1}{HW} \sum_{i=0}^{H-1} \sum_{j=0}^{W-1} \text{attn}[i,j] \right), \tag{6}$$

where $H_{\text{gt}}$ and $W_{\text{gt}}$ are grounding truth bounding box width and height, respectively. First, $P_{\text{peak}}$ checks if there exists at least one significant activation point within the target region by verifying if any pixel's attention weight exceeds a given threshold $\tau$. Second, $P_{\text{global}}$ ensures that the average attention weight within the target region is higher than the global average attention weight across the entire attention map. Together, these criteria ensure that the attention mechanism not only highlights specific points within the target area but also prioritizes the region as a whole compared to the rest of the map. The final loss function is as follows:

$$\mathcal{L}(\theta) = f(\text{attn}, gt^{\text{bbox}}, \tau) \times \mathbb{E}_t \left[ -\frac{\pi_\theta(a_t|s_t)}{\pi_{\text{old}}(a_t|s_t)} A_t + \gamma \cdot \text{KL}[\pi_{\text{old}}(\cdot|s_t), \pi_\theta(\cdot|s_t)] \right], \tag{7}$$

where $\pi_\theta(a_t|s_t)$ means the current policy, $\pi_{\text{old}}(a_t|s_t)$ means the old policy. The advantage function $A_t$ measures the relative value of an action, guiding the optimization process. Additionally, the hyperparameter $\gamma$ controls the influence of the KL divergence term, $\text{KL}[\pi_{\text{old}}(\cdot|s_t), \pi_\theta(\cdot|s_t)]$, which penalizes large deviations between the old and new policies, thereby maintaining training stability.

## 4 Experiments

### 4.1 Implementation Details

For Reinforcement finetuning, we use the QwenVL2.5-3B/7B [35] model as the base model and train ten epochs. The $\alpha$, $\beta$, $\gamma$, $\tau$ used in the formula are set as 1, 2, 0.004, and 0.2, respectively. During inference, to ensure fairness, we apply a unified and simple prompt across all benchmarks under zero-shot configurations. All experiments are conducted using 8×NVIDIA A100-80G GPUs. We evaluate our model on six benchmarks that cover grounding and agents on three different platforms, including ScreenSpot [18], ScreenSpot-v2 [7], ScreenSpot-Pro [4], AndroidControl-Low [19], AndroidControl-High [19] and OmniAct[20]. Following UGround [6], we use two commonly adopted metrics for GUI agents in evaluation: click point prediction accuracy, and step success rate, denoted as Grounding, and SR, respectively. In more detail, Grounding evaluates the performance of GUI grounding in downstream tasks. Besides, SR represents the step-wise success rate, where a step is deemed successful only if both the predicted action and its associated arguments (e.g., point for click actions and input text for type actions) are correct.

### 4.2 Experimental Results

We here evaluate our SE-GUI model by comparing it with current state-of-the-art (SOTA) models on various tasks including GUI grounding tasks, GUI low-level tasks, and GUI high-level tasks.

**Grounding capability.** We evaluate the grounding capability of SE-GUI using ScreenSpot [18], ScreenSpot-v2 [7] and ScreenSpot-Pro [4]. ScreenSpot and ScreenSpot-v2 assesses GUI grounding performance across mobile, desktop, and web platforms, while ScreenSpot-Pro focuses on high-resolution professional environments, featuring expert-annotated tasks spanning 23 applications, five industries, and three operating systems.

As shown in Tab. 1, compared to the previous SOTA model UI-TARS-72B, which is trained with large-scale data using supervised finetuning (SFT), our SE-GUI-7B achieves 8.5% improvement using only 0.2% of the data (3K vs. 14M). Furthermore, compared to the base models Qwen2.5-VL-7B and the SFT-trained Qwen2.5-VL-7B models using the same dataset, the RFT-based SE-GUI demonstrates much better performance in GUI grounding tasks. Moreover, the results in Tab. 3 reveals that SE-GUI-7B achieves 88.2% on ScreenSpot and 90.25% on ScreenSpot-v2, respectively. This highlights the effectiveness of our method in leveraging small-scale datasets to achieve significant performance improvements, which demonstrates its potential as a data-efficient and scalable approach for model training in resource-constrained environments.

Table 1: Performance comparison of different agent models across various task categories based on Text, Icon, and Average scores on ScreenSpot-Pro. Results marked in **bold** represent the best performance, and those underlined indicate the second-best performance.

| Model | CAD Text | CAD Icon | Dev Text | Dev Icon | Creative Text | Creative Icon | Scientific Text | Scientific Icon | Office Text | Office Icon | OS Text | OS Icon | Avg. Text | Avg. Icon | Avg. |
|---|---|---|---|---|---|---|---|---|---|---|---|---|---|---|---|
| *Proprietary Models* | | | | | | | | | | | | | | | |
| GPT-4o [36] | 2.0 | 0.0 | 1.3 | 0.0 | 1.0 | 0.0 | 2.1 | 0.0 | 1.1 | 0.0 | 0.0 | 0.0 | 1.3 | 0.0 | 0.8 |
| Claude Computer Use [37] | 14.5 | 3.7 | 22.0 | 3.9 | 25.9 | 3.4 | 33.9 | 15.8 | 30.1 | 16.3 | 11.0 | 4.5 | 23.4 | 7.1 | 17.1 |
| *General Open-source Models* | | | | | | | | | | | | | | | |
| Qwen2.5-VL-3B [22] | 9.1 | 7.3 | 22.1 | 1.4 | 26.8 | 2.1 | 38.2 | 7.3 | 33.9 | 15.1 | 10.3 | 1.1 | 23.6 | 3.8 | 16.1 |
| Qwen2.5-VL-7B [22] | 16.8 | 1.6 | 46.8 | 4.1 | 35.9 | 7.7 | 49.3 | 7.3 | 52.5 | 20.8 | 37.4 | 6.7 | 38.9 | 7.1 | 26.8 |
| *GUI-specific Models* | | | | | | | | | | | | | | | |
| CogAgent-18B [1] | 7.1 | 3.1 | 14.9 | 0.7 | 9.6 | 0.0 | 22.2 | 1.8 | 13.0 | 0.0 | 5.6 | 0.0 | 12.0 | 0.8 | 7.7 |
| Aria-UI [17] | 7.6 | 1.6 | 16.2 | 0.0 | 23.7 | 2.1 | 27.1 | 6.4 | 20.3 | 1.9 | 4.7 | 0.0 | 17.1 | 2.0 | 11.3 |
| OS-Atlas-7B [38] | 12.2 | 4.7 | 33.1 | 1.4 | 28.8 | 2.8 | 37.5 | 7.3 | 33.9 | 5.7 | 27.1 | 4.5 | 28.1 | 4.0 | 18.9 |
| ShowUI-2B [33] | 2.5 | 0.0 | 16.9 | 1.4 | 9.1 | 0.0 | 13.2 | 7.3 | 15.3 | 7.5 | 10.3 | 2.2 | 10.8 | 2.6 | 7.7 |
| UGround-7B [39] | 14.2 | 1.6 | 26.6 | 2.1 | 27.3 | 2.8 | 31.9 | 2.7 | 31.6 | 11.3 | 17.8 | 0.0 | 25.0 | 2.8 | 16.5 |
| UGround-V1-7B [39] | 15.8 | 1.2 | 51.9 | 2.8 | 47.5 | 9.7 | 57.6 | 14.5 | 60.5 | 13.2 | 38.3 | 7.9 | 45.2 | 8.1 | 31.1 |
| UI-R1-3B [40] | 11.2 | 6.3 | 22.7 | 4.1 | 27.3 | 3.5 | 42.4 | 11.8 | 32.2 | 11.3 | 13.1 | 4.5 | 24.9 | 6.4 | 17.8 |
| GUI-R1-3B [10] | 26.4 | 7.8 | 33.8 | 4.8 | 40.9 | 5.6 | 61.8 | 17.3 | 53.6 | 17.0 | 28.1 | 5.6 | - | - | - |
| GUI-R1-7B [10] | 23.9 | 6.3 | 49.4 | 4.8 | 38.9 | 8.4 | 55.6 | 11.8 | 58.7 | 26.4 | 42.1 | 16.9 | - | - | - |
| UI-TARS-2B [2] | 17.8 | 4.7 | 47.4 | 4.1 | 42.9 | 6.3 | 56.9 | 17.3 | 50.3 | 17.0 | 21.5 | 5.6 | 39.6 | 8.4 | 27.7 |
| UI-TARS-7B [2] | 20.8 | 9.4 | 58.4 | 12.4 | 50.0 | 9.1 | 63.9 | 31.8 | 63.3 | 20.8 | 30.8 | 16.9 | 47.8 | 16.2 | 35.7 |
| UI-TARS-72B [2] | 18.8 | 12.5 | 62.9 | 17.2 | 57.1 | **15.4** | 64.6 | 20.9 | 63.3 | 26.4 | 42.1 | 15.7 | 50.9 | 17.6 | 38.1 |
| *Ours* | | | | | | | | | | | | | | | |
| **SE-GUI-3B** | 38.1 | 12.5 | 55.8 | 7.6 | 47.0 | 4.9 | 61.8 | 16.4 | 59.9 | 24.5 | 40.2 | 12.4 | 50.4 | 11.8 | 35.9 |
| **SE-GUI-7B** | **51.3** | **42.2** | **68.2** | **19.3** | **57.6** | 9.1 | **75.0** | **28.2** | **78.5** | **43.4** | **49.5** | **25.8** | **63.5** | **21.0** | **47.3** |

Table 2: Step accuracy on AndroidControl over 500 random actions from the test split. Baseline results are from [19]. Note that previous work are trained on AndroidControl and GUIACT, while our methods in the zero-shot setting are only trained on grounding data.

| Planner | Grounding | AndroidControl_High Grounding | AndroidControl_High SR | AndroidControl_Low Grounding | AndroidControl_Low SR | OmniACT AS |
|---|---|---|---|---|---|---|
| *Supervised Setting* | | | | | | |
| GPT-4o | SeeClick [18] | - | 39.4 | - | 47.2 | 29.6 |
| GPT-4o | UGroundV1-7B [6] | - | 48.4 | - | 62.4 | 32.8 |
| *Zero-shot Setting* | | | | | | |
| GPT-4o | Qwen2.5-VL-7B [35] | 31.6 | 36.0 | 62.6 | 58.0 | 24.1 |
| GPT-4o | **SE-GUI-7B** | **65.6** | **52.8** | **79.6** | **68.2** | **34.4** |

**Agent evaluation.** We evaluate our method, SE-GUI, on three challenging GUI agent benchmarks: AndroidControl_High, AndroidControl_Low and OmniACT. The AndroidControl benchmark [19] consists of 15K tasks across 833 Android apps. Each task contains a sequence of actions grounded in screenshots and a11y trees, with both high-level intents and optional low-level step-by-step instructions. Following the evaluation protocol in [6], we randomly sample 500 steps from the test split and report step-wise accuracy (SR), where a step is considered correct only if all predicted actions and arguments match the ground truth. As shown in Table 2, SE-GUI-7B significantly outperforms all baselines across both task settings. In the low-level setting, where each step includes a fine-grained instruction, SE-GUI achieves 52.8% accuracy, outperforming GPT-4o (39.4%) and UGround-V1 (48.4%). In the high-level setting, where only the task goal is given, SE-GUI achieves 68.2%, a +5.8% improvement over the strongest baseline, UGround-V1 (62.4%). Notably, unlike prior methods such as UGround-V1 and SeeClick, which are trained on AndroidControl and GUIACT, SE-GUI is trained purely on grounding data, yet generalizes effectively to both task regimes.

Table 3: Performances on various platforms (Mobile, Desktop, Web) on **ScreenSpot** and **ScreenSpot-v2**. All experiments were conducted using raw screenshot information. Results marked in **bold** represent the best performance, and those underlined indicate the second-best performance.

| Model | ScreenSpot Accuracy (%) | | | | ScreenSpot-v2 Accuracy (%) | | | |
|---|---|---|---|---|---|---|---|---|
| | Mobile | Desktop | Web | **Avg.** | Mobile | Desktop | Web | **Avg.** |
| *Proprietary Models* | | | | | | | | |
| GPT-4o [41] | 21.9 | 17.8 | 9.4 | 18.8 | 22.5 | 22.2 | 12.4 | 20.1 |
| *General Open-source Models* | | | | | | | | |
| Qwen2-VL-7B [22] | 50.3 | 40.4 | 27.4 | 42.9 | 39.4 | 50.1 | 27.7 | 39.8 |
| Qwen2.5-VL-3B [22] | - | - | - | 55.5 | 55.5 | 44.0 | 39.1 | 46.9 |
| Qwen2.5-VL-7B [22] | - | - | - | 84.7 | 92.8 | 78.4 | 85.4 | 86.5 |
| *GUI-specific Models* | | | | | | | | |
| CogAgent-18B [26] | 57.8 | 31.6 | 40.1 | 47.4 | 50.6 | 51.6 | 54.1 | 52.8 |
| SeeClick-7B [18] | 68.1 | 48.8 | 41.8 | 53.4 | 51.8 | 65.5 | 40.7 | 53.9 |
| OSAtlas-4B [7] | 56.2 | 74.9 | 69.9 | 68.5 | 74.9 | 56.9 | 70.0 | 68.5 |
| UGround-7B [6] | 75.9 | 75.8 | 78.3 | 73.3 | 74.3 | 74.9 | 78.6 | 76.3 |
| ShowUI-2B [33] | 84.8 | 70.8 | 76.2 | 75.1 | 70.0 | 85.1 | 73.3 | 77.3 |
| OSAtlas-7B [7] | 85.0 | 78.8 | 84.5 | 82.5 | 78.3 | 85.5 | 83.8 | 83.3 |
| Aguvis-7B [8] | **86.9** | 82.4 | 84.7 | 84.4 | 89.6 | 86.8 | 84.9 | 87.3 |
| UI-TARS-2B [2] | 85.0 | 81.4 | 79.8 | 82.3 | 87.9 | 81.4 | 82.9 | 84.7 |
| *Ours* | | | | | | | | |
| **SE-GUI-7B** | 85.6 | **91.4** | **86.5** | **88.2** | **95.2** | **87.1** | **87.0** | **90.3** |

For desktop and web environments, we evaluate on the OmniACT dataset [42], which comprises 9,802 tasks spanning 38 applications and 27 websites on macOS, Windows, and Linux. Each task involves generating a complete PyAutoGUI script based on a single screenshot. We follow the DetACT pipeline for a fair comparison, but replace its multimodal detection modules with an MLLM-based grounding system. Specifically, we prompt an MLLM to generate textual descriptions of target UI elements, then use SE-GUI to predict their screen coordinates. These coordinates are composed into scripts using the original prompts and retrieval setup of DetACT, including in-context demonstrations selected by task similarity. On this benchmark, SE-GUI achieves an action score of 34.4, surpassing all other baselines, including GPT-4o (29.6) and UGround-V1 (32.8). These results demonstrate the strong generalization and grounding capabilities of SE-GUI in both mobile and desktop GUI environments, even under zero-shot settings.

## 4.3 Ablation Study

**Image resolution and data quality.** To investigate the impact of image resolution and data quality on GUI RFT, we conduct corresponding ablation experiments, with the results shown in Fig. 4(a) and Tab. 5. When training on a filtered, higher quality version of the dataset, the model achieves a 4.45% increase in accuracy. In contrast, using unfiltered, lower-quality data significantly degrades performance, reducing accuracy to just 31.31%. This highlights the importance of data quality in supporting effective model learning. We further examine the effect of image resolution by varying the maximum number of pixels from 1 million to 5 million. As the resolution increases, the model's ability to recognize small but critical UI elements can be improved, leading to steady performance gains. This suggests that higher-resolution inputs provide more detailed visual information, which is especially valuable in complex and professional GUI tasks.

Due to computational constraints, our experiments are limited to a maximum of 5 million pixels. However, the observed trend implies that further improvements in resolution may yield additional performance benefits. Overall, these results emphasize the combined importance of clean, high-quality data and high-resolution visual inputs in advancing GUI grounding capabilities.

**The ablation on hyper parameters.** As shown in Tab. 4, we observed that the relative weights of the format reward and point reward have a noticeable impact on the final performance. Specifically, setting the ratio of format reward to point reward to 1:1 leads to a slight performance drop. To address this, we adjusted the weight ratio to 1:2 during training. We attribute this to the relative simplicity of

Table 4: Ablations for SE-GUI-7B on ScreenSpot-Pro. The left table ablates the reward weights and $\gamma$, while the right table ablates $\tau$.

| Reward Weight ($\alpha : \beta$) | | $\gamma$ | | Average | $\tau$ | | | | Average |
|---|---|---|---|---|---|---|---|---|---|
| 1:1 | 1:2 | 0.04 | 0.004 | | 0.4 | 0.3 | 0.2 | 0.1 | |
| ✓ | ✗ | ✓ | ✗ | 24.85 | ✓ | ✗ | ✗ | ✗ | 42.95 |
| ✓ | ✗ | ✗ | ✓ | 27.07 | ✗ | ✓ | ✗ | ✗ | 43.87 |
| ✗ | ✓ | ✓ | ✗ | 31.31 | ✗ | ✗ | ✓ | ✗ | 44.78 |
| ✗ | ✓ | ✗ | ✓ | 34.28 | ✗ | ✗ | ✗ | ✓ | 43.51 |

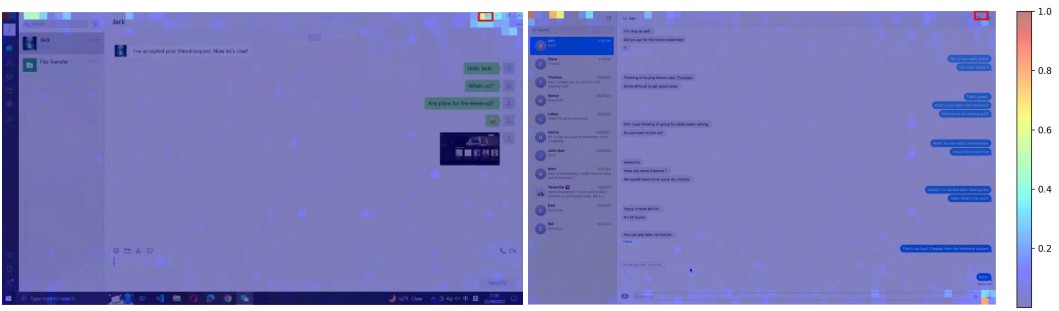

(a) Pin Jack's conversation.        (b) Initiate a video call with Ash.

Figure 3: Visualization of the model's spatial attention on different samples.

learning format consistency; reducing its weight encourages the model to focus more on prediction accuracy.

In addition, the KL divergence coefficient also influences model performance. Our experiments show that a smaller KL divergence leads to better results. We hypothesize that a large KL term forces the model to overly align with the reference policy, thereby limiting its capacity for exploratory learning. Consequently, we set the KL divergence coefficient to 0.004 in our final experimental setup.

We also conducted detailed ablation studies on the attention map filtering threshold. When the threshold is set too high, it may filter out samples with potential for reasoning. Conversely, when the threshold is too low, it fails to provide effective supervisory guidance. Therefore, we selected a threshold of 0.2.

**The effectiveness of dense point reward function.** To explore the impact of the coefficients for format rewards and accuracy rewards in the reward function on the final performance, we conduct relevant ablation experiments, as shown in Table 5. The results indicate that, compared to sparse rewards, dense rewards bring a 4.21% improvement . This is because dense rewards allow the model to receive more reward signals during the early stages of training, thus guiding the model to focus on the correct positions of UI elements.

**The effectiveness of self-evolutionary reinforcment fine-tuning.** To investigate the impact of self-evolutionary training on the final performance, we conducted relevant ablation experiments, as shown in Fig. 4(b). The results indicate that the first-stage model, trained solely on the filtered dataset SE-GUI-3k, achieved a performance of 39.97%. Subsequently, by leveraging the pre-trained model from the first stage to generate attention maps for supervising the training of the second-stage model, the performance improved by 3%. Continuing this process, where each subsequent stage utilizes the previous stage's model to generate attention maps for supervision, the third-stage model achieved a performance of 46.55%. In the fourth stage, the improvement was minimal, indicating that the model's performance had largely converged.

Furthermore, we visualize the reward curves for each training stage to better illustrate the learning dynamics. The curves are smoothed to clearly reveal the overall training trends and convergence behavior. As shown in Fig. 4(c), with the progression of training, each subsequent stage in the self-evolutionary process consistently achieves higher rewards compared to its predecessor. The final

Table 5: Ablations for data quality and reward function on ScreenSpot-Pro.

| Data quality | | reward function | | CAD | Dev | Creative | Scientific | Office | OS | **Avg.** |
|---|---|---|---|---|---|---|---|---|---|---|
| original | w/filtering | sparse | dense | | | | | | | |
| ✓ | ✗ | ✓ | ✗ | 14.6 | 26.4 | 28.6 | 40.1 | 42.6 | 14.3 | 27.1 |
| ✓ | ✗ | ✗ | ✓ | 20.3 | 30.8 | 26.1 | 38.2 | 54.3 | 25.0 | 31.3 |
| ✗ | ✓ | ✓ | ✗ | 14.8 | 31.1 | **35.8** | **50.0** | 51.3 | 25.5 | 35.8 |
| ✗ | ✓ | ✗ | ✓ | **36.4** | **34.5** | 32.3 | 47.6 | **57.8** | **29.1** | **40.0** |

stage's reward curve closely aligns with that of the preceding stage, indicating that the model has almost converged.

In addition, as shown in Fig. 4(d), we observe that on the ScreenSpot-Pro dataset, the previous state-of-the-art model, UI-TARS-72B, performs poorly in certain specialized scenarios, with accuracy dropping below 10%. In contrast, our model demonstrates strong generalization capabilities, achieving significant improvements in these challenging settings. Specifically, it boosts accuracy by nearly 40% on SolidWorks (SW) and Inventor (INV), which validates the effectiveness of our approach.

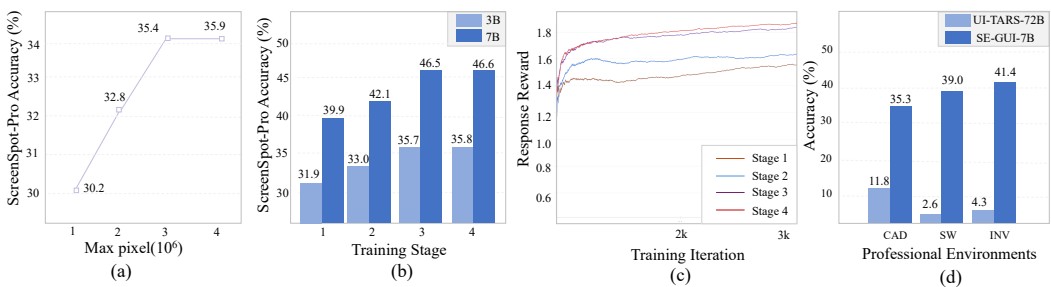

Figure 4: More experiments. (a) Perfromance of SE-GUI-3B improves as max pixel increases. (b) Performance in each training stage of self-evolutionary process. (c) The reward curves in each training stage of SE-GUI-7B in self-evolutionary process. (d) Performance comparison between UI-TARS-72B and our SE-GUI-7B in professional environments.

**Attention visualization.** We present several visualizations of the model's attention behavior. During the visualization process, we compute the attention weights from the generated tokens to these visual tokens. The attention values corresponding to the visual tokens are then mapped back to the original image resolution, yielding a spatial attention map that reflects how the model attends to different regions of the image. As shown in Fig. 3, given the user instruction "pin Jack's conversation", the model correctly attends to the upper-right region of the image. When prompted with the instruction "initiate a video call with Jack," the attention map on the right fails to localize the ground truth region, likely due to its inability to recognize the corresponding "video call" icon.

## 5 Conclusions and Limitations

**Conclusions.** In this work, we explore how to more effectively leverage reinforcement learning to unlock the potential of large multimodal models as GUI Agents. Motivated by recent studies that emphasize the importance of training data quality for RL-based methods, we employ a data filtering strategy to curate a high-quality dataset, on which we train a base model. Building upon this, we adopt a self-evolutionary reinforcement fine-tuning paradigm to progressively enhance model performance. Our approach achieves state-of-the-art results across three grounding benchmarks. We hope this study provides new insights for future research in the field of GUI Agents.

**Limitations and future work.** Due to hardware limitations, this paper explores only two model scales: 3B and 7B. For the 7B model, we further constrained the maximum input resolution to 2 million pixels to limit the number of visual tokens, which may result in the loss of fine-grained details in high-resolution images. Nevertheless, we believe that our method exhibits strong scalability and generalization capabilities, and has the potential to achieve even better performance when applied to larger models such as 32B or 72B.

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

# A GRPO preliminary

Many rule-based RL works [43; 44; 45] adopt the Group Relative Policy Optimization (GRPO) algorithm [13] for RL training. GRPO offers an alternative to commonly used Proximal Policy Optimization (PPO) [12] by eliminating the need for a critic model. Instead, GRPO directly compares a group of candidate responses to determine their relative quality.

In GRPO, given a task question, the model generates a set of $N$ potential responses $\{o_1, o_2, \ldots, o_N\}$. Each response is evaluated by taking the corresponding actions and computing its reward $\{r_1, r_2, \ldots, r_N\}$. Unlike PPO, which relies on a single reward signal and a critic to estimate the value function, GRPO normalizes these rewards to calculate the relative advantage of each response. The relative quality $A_i$ of the $i$-th response is computed as

$$A_i = \frac{r_i - Mean(\{r_1, r_2, \ldots, r_N\})}{Std(\{r_1, r_2, \ldots, r_N\})}, \tag{8}$$

where $Mean$ and $Std$ represent the mean and standard deviation of the rewards, respectively. This normalization step ensures that responses are compared within the context of the group, allowing GRPO to better capture nuanced differences between candidates. Policy updates are further constrained by minimizing the KL divergence between the updated and reference models, ensuring stable RL learning.

# B More training details.

Due to resource constraints, during training on the 7B model, we limited the maximum input resolution to 2 million pixels. We believe that increasing this limit could lead to further performance improvements. Below, we present several representative prompts used during training, along with the complete reward curves, as shown in Fig. 5.

> **Bounding box accuracy scoring prompt.**
>
> Analyze the provided cropped image from a screenshot to determine whether it contains a single, valid, and visually complete UI element.
> Criteria for validity:
> - The image must contain exactly one UI element. - The element must be entirely visible within the cropped area, with no significant cut-off parts. - The image should not consist solely of background, empty space, or meaningless fragments.
> Response format:
> Conclude with your final determination in a dedicated section:
> Conclusion Yes (if the image contains a single, valid, and complete UI element) No (if it does not meet the criteria)

> **Instruction quality scoring prompt.**
>
> Analyze whether the instruction text precisely identifies the UI element in the image based on:
> 1. Does the instruction EXACTLY match visible text/core function? 2. Could the instruction confuse similar elements in context? 3. Does it clearly indicate the required action without ambiguity?
> Instruction to evaluate: instruction
> Conclude with your final determination:
> Conclusion Yes (if all criteria are met) No (if any criterion fails)

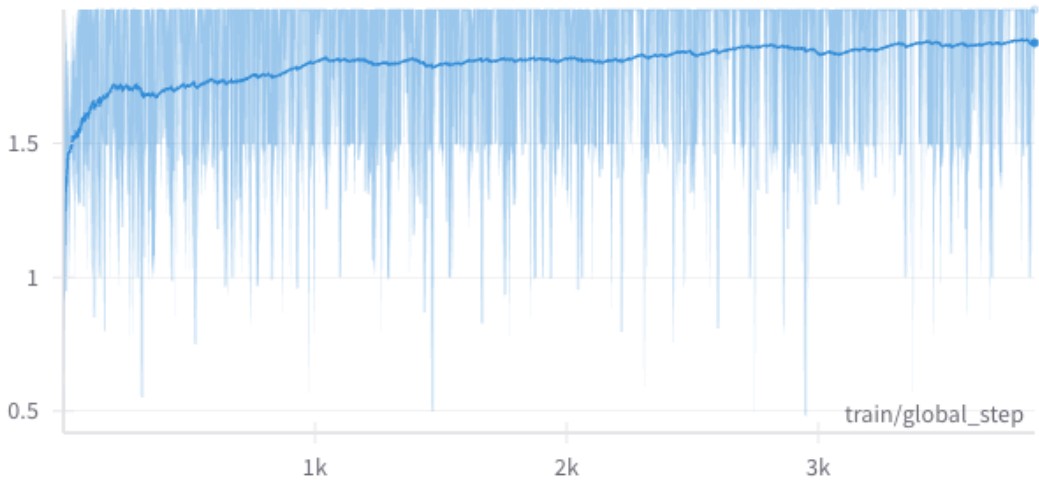

Figure 5: Training curve of our SE-GUI-7B model.

