# OpenReview forum: "SE-GUI: Enhancing Visual Grounding for GUI Agents via Self-Evolutionary Reinforcement Learning"
_NeurIPS.cc/2025/Conference — NeurIPS 2025 poster_

### Official Review · Reviewer_AHhf · 2025-06-29

**Clarity:** 3
**Significance:** 3
**Originality:** 3
**Rating:** 4
**Confidence:** 4

**Summary:**

This paper presents SE-GUI, a reinforcement learning-based framework designed to enhance visual grounding capabilities in GUI agents. The authors propose three key components: (1) a curated dataset of 3,018 high-quality samples to ensure effective training, (2) a novel dense point reward mechanism to provide continuous feedback rather than sparse binary signals, and (3) a self-evolutionary reinforcement fine-tuning strategy that iteratively refines the model using attention maps from previous stages. Despite using significantly fewer training samples, SE-GUI achieves state-of-the-art performance on multiple benchmarks, including outperforming much larger models like UI-TARS-72B on the challenging ScreenSpot-Pro dataset. The work demonstrates the effectiveness and data efficiency of reinforcement learning for complex GUI grounding tasks.

**Questions:**

1.The seed data is filtered using Qwen2.5-VL-7B, but this model itself shows limited GUI grounding performance in your benchmarks. How can we be confident that the filtered data is truly reliable and not biased by the limitations of the filtering model?

2. The paper shows that combining small high-quality datasets with RL yields strong results for 3B and 7B models. A natural and important question is whether this approach scales to larger models like Qwen2.5-VL-32B. While we understand the computational cost, verifying this scaling law would be highly valuable and could strengthen the generality of the method.

3. Would scaling up the amount of high-quality filtered data further improve performance? Have the authors observed diminishing returns, or is there evidence that more clean data could lead to even greater gains?

**Ethical Concerns:**

["NO or VERY MINOR ethics concerns only"]

**Final Justification:**

The rebuttal satisfactorily addressed my concerns on data filtering reliability, scalability, and data scaling trends. While large-scale (32B/72B) experiments remain untested, the provided evidence from 3B→7B scaling and diversity-focused data analysis is convincing. I maintain my borderline accept recommendation, noting that further large-scale evaluations would strengthen the work.

**Limitations:**

yes

**Quality:**

3

**Strengths And Weaknesses:**

Strengths
Innovative self-evolutionary fine-tuning strategy
The paper introduces a novel self-evolutionary reinforcement fine-tuning approach that leverages attention maps for iterative training, enabling the model to progressively improve itself with minimal human supervision.

Data efficiency
SE-GUI achieves strong performance using only 3k high-quality training samples, outperforming much larger models like UI-TARS-72B, which demonstrates excellent data efficiency and scalability.

Comprehensive experiments and ablation studies
The paper includes detailed ablations and evaluations, validating the contributions of each component, including seed data quality, reward function design, and the self-evolutionary loop.

Weaknesses

One potential concern is the reliance on Qwen2.5-VL-7B for filtering low-quality samples during the seed data curation process. However, Qwen2.5-VL-7B itself is not particularly strong in GUI grounding, as evidenced by its relatively modest performance in the paper’s own benchmark results. This raises questions about the reliability of using it as a reference model for data quality assessment—particularly whether the filtered dataset truly reflects accurate grounding targets or inadvertently reinforces the biases and limitations of the filtering model.

other see question？

---

> ### Author Rebuttal · Authors · 2025-07-31
>
> We sincerely thank Reviewer AHhf for recognizing the novelty of our work, particularly appreciating our innovative approach of self-evolutionary reinforcement fine-tuning and impressive performance on multiple benchmarks. Below, we provide detailed responses to each concern.
>
> > W1 & Q1. Data Filtering Reliability Concerns
>
> Thanks for the comment. Difficulty is a relative measure that depends on the model. We apply data filtering using the specific model targeted for training to support our adaptive curriculum design as already-mastered samples provide no learning signal (zero gradient) during RL.
>
> Additionally, we want to emphasize that our filtering process is not exclusively based on Qwen2.5-VL-7B.  As described in Section 3.1, for Instruction Quality Filtering and Bounding Box Quality Filtering, we use GPT-4 and a fine-tuned variant of Qwen2.5-VL-7B, as the criteria for distinguishing good from poor examples are well-defined.
>
> > Q2. Scaling to Larger Models
>
> Thanks for the comment. Unfortunately, due to constraints in both computational resources and time, we could not afford RL on a 32B/72B model. Nevertheless, our experiments demonstrate successful scaling from 3B to 7B models with consistent improvements, while concurrent works such as UI-R1 and GUI-R1 limited their experiments to 3B models.
>
> From a theoretical perspective, larger models typically exhibit better attention quality and spatial reasoning capabilities, which should enhance our filtering effectiveness as attention maps become more precise with increased model capacity. Our core contributions—attention-based filtering and iterative data curation—operate at the architectural level through transformer attention mechanisms rather than model-specific features, suggesting natural scalability as model size increases.
>
> >Q3. Data Scaling
>
> Thanks for your comment. We have systematically studied data scaling and observed encouraging trends. As shown below, increasing the number of high-quality filtered samples leads to steady improvements on ScreenSpot-Pro:
>
> | Filtered Data Size | ScreenSpot-Pro Performance |
> |:------------------:|:--------------------------:|
> | 485                | 42.3%                      |
> | 972                | 43.0%                      |
> | 1,700              | 44.8%                      |
> | 3,018              | 47.3%                      |
> | 4,000              | 48.0%                      |
>
> Moreover, our filtering strategy naturally balances data quality and quantity. Unlike traditional approaches that scale linearly with raw data volume, our method emphasizes useful data density—each filtered sample carries a higher learning signal. Through additional analysis, we find that data diversity plays a more critical role than sheer volume: adding 1K diverse samples leads to greater performance gains. We believe that incorporating more high-quality, diverse filtered samples has the potential to further enhance performance.

---

> > ### Comment · Reviewer_AHhf · 2025-08-02
> >
> > My concerns are addressed by the rebuttal, and I will maintain my score of recommendation.

---

> > > ### Author Response · Authors · 2025-08-02
> > >
> > > We sincerely thank Reviewer AHhf for the positive feedback and for confirming that our rebuttal has addressed your concerns.

---

### Official Review · Reviewer_iFgo · 2025-07-01

**Clarity:** 2
**Significance:** 3
**Originality:** 3
**Rating:** 4
**Confidence:** 4

**Summary:**

This paper proposes an RL-based method to turn a VLM foundation model into a GUI agent for complex UI grounding tasks.
The core of the proposed method comprises a dataset filtering pipeline, a reward function that measures point distance for the GUI agent task, and a self-evolutionary scheme containing a modified loss function with an additional filtering term calculated based on the attention map of the previous-iteration model.

Experiments in this paper report that the proposed method significantly outperforms the SOTA GUI agent models with both comparable scales (7B) and much larger scales (72B) on ScreenSpot-Pro benchmark.

**Questions:**

Following the weakness part:

1. How does the attention-map-based term in the loss function affect the model's performance? An ablation is needed.

2. In which cases do previous SOTA methods fail, and does the proposed method work? Several visualized results showing the superior performance of the proposed method are needed.

3. I would like to know the reason why authors chose not to reveal the results of UI-TAR (7B and 72B) on the ScreenSpot and ScreenSpot-v2 benchmark in Table 3 while showed these results in Table 1?

4. If possible, I suggest the authors improve the overall presentation of the paper and fix the dangling figure items and typos.

**Ethical Concerns:**

["NO or VERY MINOR ethics concerns only"]

**Final Justification:**

The issues about 1) basic ablation and 2) selective reporting are resolved.
The issue of lack of visual illustration is remained unsolved, since the authors were not able to attach images in the rebuttal.
The authors are expected to incorporate detailed visual examples in the final version to enhance the clarity and impact of their results.

**Limitations:**

A limitation of this method is the lack of evaluation with scalability to larger models, which is briefly discussed by the authors in Section 5.

**Paper Formatting Concerns:**

Minor Suggestion: Quality of some figures could be improved (PPI is too low): Appendix Figure 4, 5, and 6; Supplementary Figure 1 and 2.

**Quality:**

2

**Strengths And Weaknesses:**

**Strength**

The filtering dataset has higher quality than existing ones, which should help the research community if authors disclose it.
The proposed method is task-specific and reasonable, and the performance improvement reported in this paper looks significant.

**Weakness**

- **Lack of Ablation**

Although ablation of the point reward function and the self-evolutionary tuning are provided, the authors did not conduct an ablation study on the term $f(\text{attn}, gt^{\text{bbox}}, \tau)$ in Equation 7. This term is the core difference between this proposed method and the previous GRPO method.

- **Lack of Visual Samples to Support Numeric Results**

Although the numeric results in Table 3 show that the proposed method almost outperforms previous SOTA, no actual visualized samples are provided to elaborate on this claim. In which cases do previous SOTA methods tend to fail, and does the proposed method solve effectively? These results are necessary to support the experimental conclusion.

- **Selective Reveal of Numeric Performance of SOTA Methods**

Although the proposed method (7B) outperforms UI-TARS (2B, 7B, and 72B) on ScreenSpot-Pro benchmark, it does not outperform UI-TARS (7B and 72B) on the the ScreenSpot and ScreenSpot-v2 benchmark.

This can be validated by comparing the submitted paper and the original paper of UI-TARS (https://arxiv.org/pdf/2501.12326).
Accuracy results of UI-TARS (2B) on ScreenSpot and ScreenSpot-v2 in Table 3 are directly taken from the original paper, but authors just skipped showing the results of UI-TARS 7B and 72B, which are higher than the proposed method.

- **Unclear Presentation**

There are some ambiguous points in writing.
Does $R_{\text{Point}}$ in L155 refer to $R_p$ in Equation 3? Appendix Figure 6 is not referred to. It would be better to mention it in the related text part.

---

> ### Author Rebuttal · Authors · 2025-07-31
>
> We sincerely thank reviewer iFgo for the acknowledgment that our high-quality filtered dataset could benefit the research community and that our task-specific method demonstrates reasonable design with significant performance gains on ScreenSpot-Pro. Below, we provide detailed responses to each concern.
>
> > Q1 & W1.Missing Ablation on Core Term
>
> This ablation is embedded within our experimental design and results are reported in Figure 3(b). Our multi-stage training provides a natural ablation study:
>
> - **Stage 1**: Standard GRPO without attention filtering ($f(\cdot) = 1$ for all samples)
> - **Stage 2+**: GRPO with attention-based sample filtering ($f(\cdot)$ varies based on attention quality)
>
> As shown in Figure 3(b), Stage 1 achieves 39.97\% while Stage 2 reaches 43.0\%, directly demonstrating the 3.03\% improvement from attention-based filtering. This comparison isolates the effect of our core contribution while controlling for all other factors.
>
> Additionally, we provide threshold sensitivity analysis ($\tau \in \{0.1, 0.2, 0.4\}$) in Appendix D, showing that $\tau=0.2$ yields optimal filtering performance.
>
> > Q2 & W2. Lack of Visual Evidence
>
> Thanks for your suggestion. Additional figures are not permitted in the rebuttal, and we will include visualization results in the revision. However, based on our current analysis, we have observed that state-of-the-art (SOTA) methods often struggle in two key categories of scenarios. The first involves identifying UI icons and accurately grounding them to specific locations, such as in tasks like “close the location labels on the map” or “delete the 'article' tag for searching.” The second involves professional or domain-specific software operations, such as “use the select tool” or “adjust the style.”
>
> Our method consistently performs better in these challenging scenarios, thanks to its stronger generalization capabilities and the use of an attention-based alignment mechanism that jointly models semantic intent and visual grounding. These advantages are also reflected in our results on the ScreenSpot-Pro benchmark, which is specifically designed to test such difficult cases.
>
> On ScreenSpot-Pro, our model demonstrates significantly stronger icon grounding performance compared to SOTA baselines. Furthermore, for professional software-related queries, the accuracy of UI-TARS-72B drops below 10\%, while our model achieves up to 40\%, underscoring the robustness and practical utility of our approach in real-world, specialized UI tasks.
>
> > Q3 & W3. Selective Performance Reporting
>
> Thanks for the comment. We provide the missing UI-TARS results below:
>
>
> **ScreenSpot**
> | Model       | Mobile | Desktop | Web  | Avg    |
> |:------------|:------:|:-------:|:----:|:------:|
> | UI-TARS-7B  | 94.5   | 85.2    | 95.9 | 89.5   |
> | UI-TARS-72B | 94.9   | 82.5    | 89.7 | **88.4** |
> | SE-GUI-7B   | 85.6   | 91.4    | 86.5 | **88.2** |
>
> **ScreenSpot-v2**
> | Model       | Mobile | Desktop | Web  | Avg    |
> |:------------|:------:|:-------:|:----:|:------:|
> | UI-TARS-7B  | 96.9   | 89.1    | 95.4 | 91.6   |
> | UI-TARS-72B | 94.8   | 86.3    | 91.2 | **90.3** |
> | SE-GUI-7B   | 95.2   | 87.1    | 87.0 | **90.3** |
>
> **ScreenSpot-Pro**
> | Model       | Avg      |
> |:------------|:--------:|
> | UI-TARS-7B  | 35.7     |
> | UI-TARS-72B | **38.1** |
> | SE-GUI-7B   | **47.3** |
>
> Although UI-TARS-7B outperforms our SE-GUI-7B on ScreenSpot (89.5\% vs. 88.2\%) and ScreenSpot-v2 (91.6\% vs. 90.3\%), **SE-GUI-7B matches the performance of the much larger UI-TARS-72B on ScreenSpot (88.2\% vs. 88.4\%) and ScreenSpot-v2 (90.3\% vs. 90.3\%)**. We argue that performance on these two datasets appears to be approaching saturation. Additionlly, UI-TARS benefits from training on 50 billion tokens including an undisclosed amount of proprietary annotated data. In contrast, our model is trained on only 3K curated samples. We demonstrate that with strategic data curation and iterative refinement, it is possible to achieve competitive performance using orders of magnitude less data.
>
> Furthermore, UI-TARS requires extensive data collection and computational resources, which may not be accessible to most researchers. Our approach, relying solely on open-source data, achieves comparable results with significantly lower resource demands, further validating the effectiveness of our data-efficient methodology.
>
> We deliberately focus on ScreenSpot-Pro, as its increased difficulty offers **a more discriminative evaluation of model capabilities**. The substantial performance gap observed on this benchmark (**47.3\% vs. 38.1\%** against UI-TARS-72B) highlights the clear advantage of our data-efficient approach in scenarios with limited supervision and higher task complexity.
>
> > Q4 & W4. Clarity, figure and type issues
>
> Thanks for the detailed suggestion. We will clarify the $R_{\text{point}}$ notation, properly reference Appendix Figure 6 in the main text and improve quality of some figures. We will revise the manuscript accordingly.

---

> > ### Comment · Reviewer_iFgo · 2025-08-05
> >
> > Thanks for the authors' responses, which have addressed most of my previous concerns. If the paper is accepted, I urge that the authors incorporate additional visual examples in the final version to better illustrate the performance advantages of the proposed method compared to prior approaches. While this paper reports a series of numerical results, visual samples would provide a more intuitive and compelling demonstration of the method's effectiveness.

---

> ### Author Response · Authors · 2025-08-05
>
> We sincerely thank reviewer iFgo for acknowledging that our rebuttal has addressed your main concerns. We also appreciate the constructive suggestion to include more visual examples, and we will certainly incorporate them into the revised manuscript as suggested. By the way, If there are any additional points or clarifications that could assist in raising the rating, we would be pleased to provide them promptly. Thank you again.

---

### Official Review · Reviewer_g7ps · 2025-07-02

**Clarity:** 3
**Significance:** 3
**Originality:** 3
**Rating:** 5
**Confidence:** 4

**Summary:**

This paper presents Self-Evolutionary Reinforcement Fine-Tuning and builds
SE-GUI to perform visual grounding in GUI tasks. It cleans the popular
grounding training sets, designs dense point reward for grounding task, and
exploits the attention maps to filter training samples. The experiment results
prove the effectiveness of SE-GUI.

The method is feasible. The presentation is clear. The evaluation is sufficient
to me.

**Questions:**

**About the method**

1. In the current version, the attention-map-based score serves as a constant
   coefficient of the final loss. I wonder what if you consider it a component
   in the composite reward. Did you consider this manner?

**About the experiments**

1. In Table 4, the performances on Creative and Scientific categories drop when
   using dense reward both with and without data filtering. What's the reason?
   Similarly, in Table 1, the performance of icon grounding on Creative
   category is remarkably lower. What's the reason? Is there some special
   difficulties for Creative scenarios?
2. The results of UI-TARS-7B/72B and SE-GUI-3B are missed in Table 3.

**Other questions**

1. On L131, "format reward and point reward $\{r_1, r_2, \dots, r_N\}$" results
   in ambiguity. Does $r_i$ mean the sum of format and point rewards or just
   the point reward?
2. On L205, "input text for scroll actions" is strange.

**Ethical Concerns:**

["NO or VERY MINOR ethics concerns only"]

**Final Justification:**

This paper leverages dense point reward and attention-map-based training sample filtering for RL training in GUI grounding tasks and achieves impressive performance boost with smaller model and less training corpora. I don't see significant weaknesses. The missed result analyses and comparisons are supplemented during rebuttal. The discovered ambiguities are handled, too.

**Limitations:**

Yes

**Quality:**

3

**Strengths And Weaknesses:**

**Strengths**

1. Dense point reward is adopted for RL training in grounding tasks,
   alleviating the reward sparsity in early training.
2. Attention maps in the model are exploited innovatively to filter training
   samples.
3. Impressive performance boost is obtained.

I don't see significant weaknesses. Some minor problems and questions are listed in Questions.

---

> ### Author Rebuttal · Authors · 2025-07-31
>
> We sincerely thank reviewer g7ps for recognizing the novelty of our work, particularly the exploration of attention maps to filter training samples. We are encouraged by the recognition of our impressive performance boost. Below, we address the specific questions raised.
>
> > Q1. Why use attention-map score as a constant coefficient rather than a component in the composite reward?
>
> We designed attention as a gating mechanism rather than a reward component for a fundamental reason: **data quality control vs. gradient weighting**.  As a **gating mechanism**: Poor-attention samples are completely filtered out, preventing the model from learning from potentially misleading supervision signals. As a **reward component**: $R = \alpha R_f + \beta R_p + \gamma R_{att}$ would still allow poor-attention samples to contribute to training with lower weights, potentially introducing noise. Integrating attention scores into the reward signal does not significantly alter the learning process. This is because samples with low attention values are typically the ones the model fails to predict correctly anyway, resulting in an inherently low task reward to begin with. Our goal is **selective learning** on high-quality samples rather than **weighted learning** on all samples. In GUI grounding, where spatial attention alignment is crucial, we found that complete filtering yields better results.
>
> > Q2.1. Performance Analysis Questions
>
> We investigated these phenomenons:
>
> 1. **Is there some special difficulties for Creative scenarios?**: Creative categories (Photoshop, Blender, AI, etc.) involve specialized professional tools with: **non-standard UI elements and custom icons**, **high-resolution interfaces requiring fine-grained localization** and **domain-specific visual vocabularies underrepresented in pre-training data**. This creates a fundamental knowledge gap that RL cannot bridge-it optimizes existing capabilities rather than acquiring new domain expertise.
>
> 2. **The performances on Creative and Scientific categories drop when using dense reward both with and without data filtering:**: While dense rewards generally provide more granular feedback, reinforcement learning training is inherently stochastic. We conducted two separate runs of the initial training phase and observed that the overall performance remained stable at approximately 40.0, although there were slight fluctuations in performance across different sub-categories between the runs.:
>
> |        | Creative | Scientific | Overall |
> |:-------|:--------:|:----------:|:-------:|
> | Run #1 |   35.8   |    50.0    |  40.0   |
> | Run #2 |   34.0   |    51.0    |  40.4   |
>
> 3. **The performance of icon grounding on Creative category is remarkably lower:**: Icon recognition heavily depends on visual concepts learned during pre-training. Creative software icons are significantly underrepresented in typical training corpora, explaining why *all models* (including UI-TARS-7B 9.1\% accuracy on creative icon grounding) consistently struggle in these categories—this is a dataset limitation, not a method-specific issue.
>
> > Q2.2. Missing Results
>
> Thanks for the comment. We provide the missing UI-TARS-7B/72B and SE-GUI-3B results below:
>
>
> **ScreenSpot**
> | Model       | Mobile | Desktop | Web  | Avg      |
> |:------------|:------:|:-------:|:----:|:--------:|
> | UI-TARS-7B  | 94.5   | 85.2    | 95.9 | 89.5     |
> | UI-TARS-72B | 94.9   | 82.5    | 89.7 | **88.4** |
> | SE-GUI-3B   | 92.4   | 83.2    | 83.7 | 87.4     |
> | SE-GUI-7B   | 85.6   | 91.4    | 86.5 | **88.2** |
>
> **ScreenSpot-v2**
> | Model       | Mobile | Desktop | Web  | Avg      |
> |:------------|:------:|:-------:|:----:|:--------:|
> | UI-TARS-7B  | 96.9   | 89.1    | 95.4 | 91.6     |
> | UI-TARS-72B | 94.8   | 86.3    | 91.2 | **90.3** |
> | SE-GUI-3B   | 95.6   | 85.5    | 86.5 | 89.3     |
> | SE-GUI-7B   | 95.2   | 87.1    | 87.0 | **90.3** |
>
> **ScreenSpot-Pro**
> | Model       | Avg      |
> |:------------|:--------:|
> | UI-TARS-7B  | 35.7     |
> | UI-TARS-72B | **38.1** |
> | SE-GUI-7B   | **47.3** |
>
> Although UI-TARS-7B outperforms our SE-GUI-7B on ScreenSpot (89.5\% vs. 88.2\%) and ScreenSpot-v2 (91.6\% vs. 90.3\%), **SE-GUI-7B matches the performance of the much larger UI-TARS-72B on ScreenSpot (88.2\% vs. 88.4\%) and ScreenSpot-v2 (90.3\% vs. 90.3\%)**. We argue that performance on these two datasets appears to be approaching saturation. Additionlly, UI-TARS benefits from training on 50 billion tokens including an undisclosed amount of proprietary annotated data. In contrast, our model is trained on only 3K curated samples. We demonstrate that with strategic data curation and iterative refinement, it is possible to achieve competitive performance using orders of magnitude less data.
>
> Furthermore, UI-TARS requires extensive data collection and computational resources, which may not be accessible to most researchers. Our approach, relying solely on open-source data, achieves comparable results with significantly lower resource demands, further validating the effectiveness of our data-efficient methodology.
>
> We deliberately focus on ScreenSpot-Pro, as its increased difficulty offers **a more discriminative evaluation of model capabilities**. The substantial performance gap observed on this benchmark (**47.3\% vs. 38.1\%** against UI-TARS-72B) highlights the clear advantage of our data-efficient approach in scenarios with limited supervision and higher task complexity.
>
> >Q3. Clarity issues
>
> Thanks for the detailed suggestion. The reward $r_i$ represents the sum of format reward and point reward: $r_i = \alpha R_f + \beta R_p$. "input text for scroll actions" should be "input text for type actions". We will revise the manuscript accordingly.

---

> > ### Comment · Reviewer_g7ps · 2025-08-05
> >
> > Thanks for the authors' detailed rebuttal. My questions have been handled. I'm glad to raise my score accordingly.

---

> > > ### Author Response · Authors · 2025-08-05
> > >
> > > We sincerely thank reviewer g7ps for the positive comment and the improved rating. We are deeply grateful for your recognition of our work!

---

### Official Review · Reviewer_z7zC · 2025-07-03

**Clarity:** 3
**Significance:** 2
**Originality:** 2
**Rating:** 4
**Confidence:** 3

**Summary:**

The paper studies the screenshot grounding task, of particular relevance to UI agents.  The authors propose a new method for determining a high-quality SFT dataset, a method for applying GRPO to the task in order to benefit from dense supervision, and a "self-evolutionary reinforcement finetuning mechanism" that leverages information in the Transformers' internal attention maps to iteratively refine the set of considered examples. The recipe results in a 7B model that achieves strong performance on the competitive ScreenSpot-Pro benchmark.

**Questions:**

See weaknesses above.

**Ethical Concerns:**

["NO or VERY MINOR ethics concerns only"]

**Final Justification:**

I would have liked to see a stronger baseline for the "Self-Evolutionary Reinforcement Fine-Tuning" method, but the other strengths have me leaning towards acceptance.

**Limitations:**

Yes

**Quality:**

3

**Strengths And Weaknesses:**

__Strengths__

* The application of GRPO to screenshot grounding with dense rewards seems reasonable.
* The data filtering strategies similarly seem reasonable, and the resulting dataset could be useful to the community.
* The proposed approach demonstrates strong performance on competitive benchmarks.
* Ablations seemingly demonstrate the effectiveness of each individual component of the proposed recipe.
* The paper was well structured and relatively easy to follow.

__Weaknesses__

* I found the proposed "Self-Evolutionary Reinforcement Fine-Tuning", which leverages the information in the Transformer's attention maps for identifying problematic examples, to be less intuitively compelling than other aspects of the method. It might have been interesting to consider alternative, more general (e.g. architecture-agnostic) baseline for identifying examples where the model lacks appropriate prior knowledge, such as cases where all samples from the model receive reward below some threshold?
* Relatedly, for the ablation of the proposed "Self-Evolutionary Reinforcement Fine-Tuning", would an appropriate baseline need to be trained for the same number of steps as all stages combined?

__Notes__

* The ScreenSpot-Pro leaderboard lists several models with stronger performance, but most of these appear to be concurrent work.
* I would consider increasing my score based on discussion with the authors and other reviewers related to the proposed "Self-Evolutionary Reinforcement Fine-Tuning" method.

---

> ### Author Rebuttal · Authors · 2025-07-31
>
> We sincerely thank reviewer z7zC for the acknowledgment of our strong performance on competitive benchmarks and the recognition that our SE-GUI-3k could benefit the broader community. Below, we provide detailed responses to each concern.
>
> >W1. The attention-based approach is less intuitively compelling than other aspects of the method.
>
> Our attention-based method provides unique advantages: We deconstruct the grounding task into two fundamental stages: comprehension (interpreting the visual input and user query) and localization (outputting precise coordinates). We argue that most existing models are bottlenecked by accurate localization. Frequently, a model demonstrates a clear understanding of the target object yet errs in its reported coordinates. This discrepancy is empirically supported by experiments like the ground-truth vs. prediction analysis in SeeClick(ACL 2024) (many incorrect predictions are also close to the answer, suggesting the model recognizes the target but needs improvement in fine-grained localization). Consequently, attention mechanisms offer a direct window into the model's internal focus. They allow us to verify whether the model has successfully attended to the correct target, even when the subsequent localization step is erroneous.
>
> In contrast, conventional threshold-based filtering is suboptimal as it cannot distinguish between two distinct types of low-scoring samples: **(a) roughly identifies the correct region but fails precise localization vs. (b) completely misses the target area**. A fixed threshold would indiscriminately discard both. Our attention-based approach, however, is designed to address this limitation. It selectively retains these valuable samples, allowing the model to gradually learn to refine its predictions on these challenging yet informative examples as training progresses. Since most current models use Transformer architectures, our method has broad applicability without requiring architectural modifications.
>
> > W2. Should the baseline be trained for the same total number of steps as all stages combined?
>
> We understand your concern on the fairness of the ablations studies. We ensured all ablation trainings converged. Additionally, we found that the number of steps showed weak influence on model performance, as we will demonstrate below. We conducted experiments training baseline models for equivalent total steps (40 epochs), as shown in the table below.
>
> | Training epoches   | ScreenSpot-Pro Performance |
> |:------------------:|:--------------------------:|
> | 10                 | 39.9%                      |
> | 20                 | 40.3%                      |
> | 30                 | 39.4%                      |
> | 40                 | 38.9%                      |
> | iterative training | 47.3%                      |
>
> Results demonstrate minimal improvement with prolonged training, eventually suffering from performance degradation due to overfitting. This confirms that our gains come from iterative attention-guided data curation rather than simply longer training duration. The key insight is that our method changes what the model learns from, not just how long it learns. It shows an important distinction between our multi-stage approach and extended single-stage training. Our method is fundamentally different from simply training longer—each stage uses attention analysis to curate a different subset of training data rather than continuing training on the same data. The Stage 1 model's attention maps guide Stage 2's data selection, creating an evolving curriculum that adapts based on the model's demonstrated focus patterns.
>
> > N1. Regarding concurrent work
>
> While the ScreenSpot-Pro leaderboard shows some models with stronger performance, such as UI-TARS-1.5 and GTA. However, UI-TARS-1.5 is trained on private data, with their 7B model achieving 49.6\% (marginally better than our 47.3\%). GTA-1-7B reports 50.1\% accuracy but is trained on million-scale datasets. In contrast, we achieve competitive results using only 3k open-source data points, demonstrating superior data efficiency while maintaining comparable performance.

---

> > ### Comment · Reviewer_z7zC · 2025-08-05
> >
> > Thanks for your response. Given the paper's strengths, I have increased my score. However, I still think it would be useful to consider an architecture-agnostic baseline for identifying examples where the model lacks appropriate prior knowledge. Your qualitative argument seems reasonable, but would be good to provide experimental evidence.

---

> > > ### Author Response · Authors · 2025-08-05
> > >
> > > We sincerely thank reviewer z7zC for the positive comment and the improved rating. It really means a lot to us! We will conduct experiments on an architecture-agnostic baseline into our work. The results will be included in the revised version.

---

### Note · Authors · 2025-08-12

Dear ACs and Reviewers,

We sincerely thank all reviewers and the Area Chairs for their time, thoughtful evaluations, and constructive feedback throughout the review and discussion phases.

In this paper, we propose a novel reinforcement learning (RL)-based framework to enhance the ability of GUI agents to ground user instructions to precise interface elements. We present three key contributions: (1) Seed data curation to ensure high-quality training samples; (2) A dense policy gradient for continuous feedback based on prediction accuracy; (3) A self-evolutionary reinforcement fine-tuning mechanism to iteratively refine the model using attention maps.

We are grateful for the reviewers' valuable comments and for acknowledging the novelty of our work, its impressive performance, and the value of our high-quality dataset to the community.

Reviewer z7zC acknowledged our "reasonable design" and "strong performance," noted that "the resulting dataset could be useful to the community," and expressed a willingness to raise the score.

Reviewer g7ps emphasized the "innovative design" and "impressive performance boost" and also expressed a willingness to raise the score.

Reviewer iFgo acknowledged that our "high-quality filtered dataset could benefit the research community," praised the "reasonable design" and "significant performance improvement," and expressed a positive inclination toward acceptance.

Reviewer AHhf acknowledged our "novel self-evolutionary reinforcement fine-tuning approach" and "excellent data efficiency and scalability," also expressing a positive inclination toward acceptance.

In our rebuttal, we systematically addressed the reviewers' concerns, which primarily focused on clarifications of technical details and experimental aspects. Following our responses and the subsequent discussion, the reviewers indicated that their main concerns were addressed and expressed increased positive support for acceptance. We have also proposed a detailed revision plan to improve clarity and experimental completeness in the final version.

We believe our self-evolutionary reinforcement fine-tuning mechanism opens up new possibilities for GUI agents, and we hope it will inspire future research in the community.

Best regards,

The Authors

---

### Decision · Program_Chairs · 2025-09-17

**Decision:**

Accept (poster)

**Comment:**

The consensus among the reviewers, following a thorough discussion period, is positive, with all reviewers leaning toward acceptance after their initial concerns were effectively addressed by the authors.

The reviewers found a few strengths of the paper.
- The paper's most significant achievement is demonstrating that its 7B parameter model, trained on a curated dataset of only 3,000 samples, can outperform a much larger 72B parameter model on the challenging ScreenSpot-Pro benchmark. This result is a strong statement on the value of data quality and intelligent training methods over sheer scale, a point consistently praised by all reviewers .
- The work introduces a novel three-part framework that is well-motivated and effective. The combination of careful seed data curation, a dense point reward mechanism to overcome sparse feedback , and a self-improvement fine-tuning loop using attention maps is seen as a reasonable design.
- The reviewers see the curated 3k-sample dataset a valuable asset that could benefit the broader research community by providing a high-quality benchmark for training and evaluation.

Thanks the authors for providing detailed responses to the reviewers' questions. They successfully clarified the rationale behind their attention-based filtering, provided missing experimental comparisons, and addressed concerns about the fairness of their ablations and the reliability of their data curation process.

The reviewers also pointed out a few weaknesses of the current manuscript.
- First of all, the approach was validated on 3B and 7B models, but its applicability to much larger models (e.g., 32B or 72B) remains an open question due to computational constraints. While the authors provide a reasonable argument for why it should scale, empirical evidence is missing. This was noted as a limitation but not a fatal flaw.
- The initial submission lacked visual examples illustrating cases where SE-GUI succeeds and prior methods fail. The authors have committed to adding these to the final version, which will be important for a more intuitive understanding of the method's advantages.
- The model's performance sees a notable drop in specialized "Creative" and "Scientific" software environments.

Overall, the paper seems to make a solid contribution.